# Cerebellar and Striatal Implications in Autism Spectrum Disorders: From Clinical Observations to Animal Models

**DOI:** 10.3390/ijms23042294

**Published:** 2022-02-18

**Authors:** Mathieu Thabault, Valentine Turpin, Alexandre Maisterrena, Mohamed Jaber, Matthieu Egloff, Laurie Galvan

**Affiliations:** 1Laboratoire de Neurosciences Expérimentales et Cliniques, Institut National de la Santé et de la Recherche Médicale, Université de Poitiers, 86073 Poitiers, France; mathieu.thabault@univ-poitiers.fr (M.T.); valentine.turpin@univ-poitiers.fr (V.T.); alexandre.maisterrena@univ-poitiers.fr (A.M.); mohamed.jaber@univ-poitiers.fr (M.J.); matthieu.egloff@univ-poitiers.fr (M.E.); 2Centre Hospitalier Universitaire de Poitiers, 86021 Poitiers, France

**Keywords:** ASD, cerebellum, striatum, epigenetics, motricity, autism

## Abstract

Autism spectrum disorders (ASD) are complex conditions that stem from a combination of genetic, epigenetic and environmental influences during early pre- and postnatal childhood. The review focuses on the cerebellum and the striatum, two structures involved in motor, sensory, cognitive and social functions altered in ASD. We summarize clinical and fundamental studies highlighting the importance of these two structures in ASD. We further discuss the relation between cellular and molecular alterations with the observed behavior at the social, cognitive, motor and gait levels. Functional correlates regarding neuronal activity are also detailed wherever possible, and sexual dimorphism is explored pointing to the need to apprehend ASD in both sexes, as findings can be dramatically different at both quantitative and qualitative levels. The review focuses also on a set of three recent papers from our laboratory where we explored motor and gait function in various genetic and environmental ASD animal models. We report that motor and gait behaviors can constitute an early and quantitative window to the disease, as they often correlate with the severity of social impairments and loss of cerebellar Purkinje cells. The review ends with suggestions as to the main obstacles that need to be surpassed before an appropriate management of the disease can be proposed.

## 1. Introduction

Autism spectrum disorder (ASD) is a neurodevelopmental condition manifested by early onset of (i) persistent deficits in communication and social interactions and (ii) restricted patterns of behaviors, activities or interests (DSM5). The severity range of each symptom, and the fact that other abnormalities can accompany ASD, led to the “spectrum” umbrella term. Comorbidities vary among patients, but the most common ones are: sleep disorders (up to 80%), intellectual deficits (45%), epilepsy (up to 30%) and, interestingly, motor abnormalities (79%) [1]. The WHO considers that ASD affects 1 in 160 children worldwide following a ratio of 3 boys for 1 girl [2], suggesting male susceptibility. To this day, the diagnosis age ranges from infant to adult, depending on the severity of the symptoms and the social environment.

The etiology of ASD is still uncertain, but evidence has strongly linked genetic, epigenetic and environmental factors to it. The gathered knowledge in this domain is interdependent with technological advances. The possibility to collect and analyze broad genetic information for the past ten years has led to breakthroughs. The first pieces of evidence of genetic involvement in ASD were the case studies of monozygotic twins. These twins displayed a 60% probability to share the same ASD diagnosis, whereas dizygotic twins’ probability was at 10% [3,4,5]. More importantly, the incidence of having an ASD child is proportional to the child’s shared genomic percentage with an ASD parent or sibling [6,7,8]. The advancement of sequencing technics led to the possibility of analyzing the patient’s whole genome, which helped identify more than 1200 susceptibility genes (SFARI gene, https://www.sfari.org/resource/sfari-gene/, accessed on 14 September 2021). These genes can be broadly categorized into two families, involving either chromatin modeling or synapse formation [9].

Perinatal factors have been identified as increasing ASD incidence, including maternal infection and maternal toxin exposure. A 2016 meta-analysis of more than 40,000 ASD cases highlighted an increased risk of ASD in children whose mother had a viral infection during pregnancy, leading to hospitalization [10]. One hypothesis underlying this observation is that maternal immune regulation leads to increased production of inflammatory cytokines. There is not a consensus about how the cytokines could reach the fetus brain. They could reach it by disrupting the brain–blood barrier (BBB) [11] or by the actions of proinflammatory cytokines able to cross the BBB (for review, see [12]). Several pharmacological agents are also responsible for increasing ASD incidence in children exposed during prenatal development. For instance, valproic acid (VPA) is widely used as an antiepileptic drug and a mood stabilizer. It has various effects, from inhibiting the histone deacetylase [13] to the GABA signaling potentiation [14]. Pregnant women under VPA medication are four to five times more likely to have an ASD child even when VPA was taken at the lowest doses [15] (for review, see [16] and Table 1). We will focus on the VPA as an environmental model of ASD in this review event, though; others have been described as the phthalic acid ester exposition (for review, see [17]).

Some of the mutations and perinatal factors increasing ASD incidence have led to the generation of corresponding animal models, with some showing good face validity compared to clinical conditions. However, not all ASD symptoms can be reproduced in rodents. For example, while social interaction and motor stereotypy can be robustly observed and scored in rodents using various and complementary behavioral tests, cognitive stereotypy and intellectual disability cannot be adequately assessed. Speech deficits are also a symptom in patients, which cannot be directly translated in the mouse models. The closest feature that can be measured is ultrasonic vocalizations (USVs). USVs are used by pups separated from their mother or littermates, by juveniles when playing and by adults during social interactions or mating. Although their exact significance is unclear, many different USVs parameters were modified in ASD mice compared to wild type. Shank1^−/−^ pups emitted less isolation-induced USV with a shift in frequencies than control mice [18], but not Shank3^−/−^ pups, where other parameters were modified [19]. Additionally, no difference in the number of USVs emitted by pups was found between the Frm1^−/−^ mice and wild type but rather a change in specific frequency [20]. Thus, it seems challenging to observe consistent and robust USV changes through different ASD mouse models.

Since 1943 and the first description of autistic features by Leo Kanner [21], followed almost concomitantly in 1944 by Hans Asperger [22], motor and cognitive impairments have been within the core of ASD symptoms. Currently, studies focus more on cognitive aspects of ASD and the corresponding physiopathology and brain regions [23,24]. The notion linking ASD to a focal cerebral dysfunction has drastically evolved toward a consensus of cerebral multi-regional reorganization during development. We will focus this review on two complex brain areas involving sensory and motor functions: the cerebellum and the striatum.

## 2. Cerebellar Involvement in ASD

### 2.1. Structure and Function of the Cerebellum

All vertebrates have a cerebellum, although not all have a cortex [34]. The cerebellum might have first emerged in fish, providing computational analysis of sensory feedback from their lateral lines and hair cells on their skin that help them detect their environment [35]. The primary role of the cerebellum has long been linked to coordination and control of movement. Its implication in higher function has first been proposed by Leiner, following the observation in higher primates of an enlargement of the cerebellum that paralleled that of the frontal cortex during phylogenic evolution [36]. A few years later, Middelton and Strike provided anatomical evidence of cerebellar and basal ganglia involvement in higher cognitive function [37]. Since then, many studies highlighted the eminent role of the cerebellum in cognitive and emotional function, and that was attributed to its connections with cortical and subcortical centers [38,39]. In addition, extensive connections between the cerebellum and frontal associative cortical areas suggest a critical role in treating sensory motor information from visual, auditive and sensory input, as well as a role in memory, language processing and planification [40].

Although relatively simple in its general structure, as well as afferent and efferent projections and cellular organization, the anatomy of the cerebellum can be difficult to describe. This is mainly due to three factors: (i) the structure/function of the cerebellum is quite different depending on whether one is considering the anteroposterior or parasagittal plans; (ii) each part of the cerebellum can have several names depending on the user and the current trend; (iii), not all authors agree on what each part encompasses. When viewed from the anteroposterior axis, the cerebellum is formed from three lobes: the vestibulocerebellum, the posterior cerebellum and the anterior cerebellum, which are separated by deep fissures. Each lobe is constituted by lobules numbered I to X, giving the cerebellum its folded shape. The vestibulocerebellum (also called lobule X) comprises a median nodule and flocculus lateral ones. It is directly connected to the vestibular nuclei within the brainstem and receives projections from the auditory nerve. Most of the efferent projections from the cerebellum originate from four deep cerebellar nuclei within the vestibulocerebellum that receive projections from collateral fibers, parallel fibers and Purkinje cells (PC). This is the oldest part of the cerebellum phylogenetically, and it is shared among all vertebrates. It handles perception of self-motion and spatial orientation through the auditory system of the head’s position and motion. The anterior lobe is composed of lobules I–V and includes the medial vermis part. It receives proprioceptive information from the body and limbs, as well as from the visual and auditive systems. It sends projections toward the cortex and the spinal cord, modulating the descending motor system. The anterior lobe is responsible for balance, posture and consequent movement adjustments in time and space. The posterior lobe is composed of lobules VI–IX. This lobe constitutes the cerebellar hemispheres and is the most recent and evolved part of the cerebellum. It is only present in higher mammals and is particularly enlarged in the human species. The posterior lobe receives projections exclusively from the cerebellar cortex and sends projections to the ventrolateral part of the thalamus. It is thought to be implicated in movement planification, motor learning, timing, language processing and emotional, cognitive and social functions.

From a parasagittal perspective, the central part of the cerebellum is constituted by the vermis, followed by the paravermis and ending laterally by the cerebral hemispheres, which constitute the most significant part in the primate’s cerebellum. Each lobule can have a different function and set of projections, depending on its parasagittal plan. Of interest to us here are crus I and crus II that constitute the lateral parts of lobules VI and VII, respectively, within the posterior lobe and that are affected in ASD. Crus I and crus II are homologous in human and non-human primates. However, rodent crus I corresponds to crus I and crus II in primates [41], while rodent crus II may be homologous to HVIIB.

At the cellular level, the cerebellar cortex is organized into three layers that are, from the surface to the white matter: (i) the molecular layer, a cell-poor layer containing stellate and basket cells as well as PC dendrites connected to parallel fiber axons from granule cells and to climbing fibers. The climbing fiber axons are also found in the granular layer, (ii) the PC layer, a thin single cell layer and (iii) the granule cell layer containing what is thought to constitute more than 50% of whole nerve cells within the brain (up to 10^11^ neurons) [42]. Granule cells receive projections from mossy fibers originating from the vestibular nuclei. In turn, they project excitatory parallel fibers to PC dendrites within the molecular layer, each parallel fiber contacting 100 PC. Each PC receives an approximate number of 150,000 synapses from parallel fibers and projections emanating from a single climbing fiber [43] originating from the inferior olive. The PC are large neurons with flat and rich dendritic arborization. Their inhibitory GABAergic axon projects through the granular layer to the vestibular nuclei and constitutes the sole efferent projection from the cerebellar cortex (Figure 1).

The various functions of the cerebellum that have been described to date, whether motor or cognitive, seem to converge toward a peculiar capacity of the cerebellum to estimate and keep track of time. James Albus proposed that the anatomy and physiology of the cerebellum both point toward a pattern-recognition data processing that allows the handling and storage of information based on trial (intent) and error (action) and that this is achieved by weakening the synaptic weights rather than by strengthening them [36,57]. Clumsiness and deficits in motor coordination and manual dexterity, abnormal balance gait and posture are all dependent on the cerebellar function and are affected in ASD [58,59,60]. These deficits can be detected even in the first months of life, with affected babies exhibiting difficulties positioning their body when carried, hypotonia and uncoordinated movements.

Based on the existing knowledge of the sensory motor role of the cerebellum and suspecting its implication in psychiatric disorders, Dow and Moruzzi developed test batteries early in 1958 on patients with various psychiatric disorders. They reported severe cerebellar-type impairments in what was referred to as patients suffering from autistic and Asperger syndromes [61]. Since then, a growing number of papers have reported substantial implications of the cerebellum in motor and non-motor deficits in ASD. From this aspect, the severity of cerebellar injury in premature infants is predictive of the severity of ASD symptoms in adult age [62]. Indeed, there are three cerebellar-related deficits reported in ASD patients based on imaging and post-mortem histological observations: (i) a decrease in the number of PC, (ii) reduced cerebellar volume and (iii) disrupted circuitry between the cerebellum and connecting brain areas, such as the thalamus, the pons and the cortex [63,64]. As PC are inhibitory, this, in turn, leads to hypersensitivity of cerebellar target areas [65,66].

### 2.2. Anatomical Evidence of Cerebellar Involvement in ASD

The involvement of cerebellum in the ASD context is also supported by genetic evi-dence. Indeed, 38 genes, whose expression is enriched in the cerebellum, were identified as ASD-linked susceptibility genes [25]. Several studies showed that ASD patients displayed hypoplasia (−12% [67,68], for review, see [63]). Using magnetic resonance imaging (MRI) in both adult ASD and healthy patients, Murakami et al. reported a reduced size of both vermis and hemispheres in ASD patients without and with mild mental retardation [67]. This alteration is not always found in ASD mouse models and could be due to the spectrum of the disease or the heterogeneous way of assessing it at the levels of (i) methodology (stereology vs. MRI), (ii) diversity of genetic background of ASD animals, (iii) animal age and (iv) gender (for review, see [69]). A decrease in cerebellar volume has also been reported in multiple genetic pathologies related to ASD, including fragile X syndrome (FXS) [70,71], Rett syndrome [70,71,72], the Phelan–McDermid syndrome, including *Shank3* deletion [73,74] (SH3 and multiple ankyrin repeat domains 2), and *NLGN4* (Neurologin-4) associated non-syndromic X-linked ASD [46]. Decreases in cerebellar volume can also be observed in corresponding animal models, such as the Fmr1 Knock-out (KO) [44] and Nlgn4 KO [75] mice models (Table 2).

PC loss has been consistently described in ASD patient brains, and cerebellar hypoplasia was found in most cases [49,62,63,75]. In our previous articles, three ASD mice models (Shank3^∆C/∆C^ [49], VPA [47] and polyinosinic:polycytidylic acid (poly I:C) [48]) displayed no global changes in cerebellum size, although they did show PC loss predominantly in crus I and II. Beyond the cell count, cerebellar connectivity is also affected in ASD, and white and gray matter abnormalities have been observed by voxel-based morphometry. Analysis of cerebellar white matter in low- and high-functioning ASD young male children shows a significant increase in white matter volume compared to a typical children group [77]. That characteristic has been robustly used as a prediction for ASD diagnosis in this same study. A specific change in cerebellar white matter is typical in 2–3-year-old ASD children, which is not observed in older ASD children and adolescents [78]. A few studies have investigated fiber tracts and myelin differences in ASD patients using diffusion tensor imaging (DTI), revealing changes in white matter structure and projections. The evaluation of white matter in male ASD children (6–12 years old) shows a thickening of the left cerebellar peduncle and of both middle cerebellar peduncles compared to typical children [76]. Interestingly, motor-related structures were also affected, such as the left putamen and the corticocortical pathway [76]. Considering motor deficits in ASD, another study assessed, contingently, motor function and DTI in ASD children (5–14 years old, twelve males and one female) compared to a non-ASD control age group [100]. In this study, ASD children displayed the poorest motor function scores (manual dexterity, ball skills and balance) compared to neurotypical children. Additionally, these motor deficits were correlated with a decrease in fractional anisotropy (white matter microstructural integrity index) bilaterally in the superior cerebellar peduncle [100], the efferent cerebellar tract to the midbrain being mainly composed of the cerebellothalamic tract. Taken together, these white matter analyses reveal an alteration of cerebellar motor pathways in ASD.

Lateralization of functional connectivity patterns is modified in ASD, revealing changes in functional topography. Noonan et al. assessed the functional connectivity of ASD brains by asking adult patients to perform a memory task while undergoing an MRI. They selected high functioning ASD adult males who showed the lowest score at general word recognition and source recognition performance compared to a control group. Although ASD and control groups showed no differences in functional connectivity in the left hemisphere, ASD patients showed greater connectivity in the right hemisphere, including the right supplementary motor areas and the cerebellum [101]. This right overconnected lateralization of the ASD cerebellum was also observed in ASD children and adolescents [81]. Interestingly, the performance of a simple motor task (self-paced button press with the dominant thumb) by young ASD adults and control patients while being imaged revealed that the magnitude of activation of the ipsilateral anterior cerebellum is strongly increased in ASD patients [68]. Moreover, the authors found that contralateral and posterior cerebellar regions that are not generally associated with simple motor tasks were abnormally active in ASD patients [68].

Consequently, based on the over-recruitment of crus I and II in ASD patients, as well as in cerebral regions involved in both cognitive and motor tasks, these regions might be essential for understanding the behavior impairments in ASD patients. From this aspect, MRI analysis in young children showed that ASD children with repetitive movements had a negative correlation with cerebellar vermis area of lobule VI and VII (crus I and II) [82]. Evidence from 10-year-old ASD children revealed higher functional connectivity between the right crus I and the left inferior parietal lobule than neurotypical children [102]. This connectivity is not the only one impaired in ASD. Similar observations were reported regarding projections originating from the right crus I to the left mPFC in ASD patients [79] and in 30 out of 94 ASD mice models [83]. Consequently, chemogenetics-specific inhibition of PC in the right crus I in the tuberous sclerosis complex 1 (Tsc1) ASD mouse model led to increased firing in the left mPFC associated with improved sociability behavior, inflexibility and motor stereotypies [80]. This pathway is polysynaptic, as it anatomically relays on the ventral medial thalamus. In the Tsc1 ASD mouse models, the optogenetic inhibition of these specific thalamic neurons was sufficient to prevent both social impairments and repetitive behavior [83].

The posterior cerebellar circuits allow the processing and integration of multisensory information, including visual, proprioceptive and somatosensory, which may be reduced in ASD patients. This raises the hypothesis of compensating mechanisms in ASD patients regarding their sensory forward control deficits in prehension, gait and postural control. When high-functioning ASD children have been asked to perform a simple motor task, such as a “grip”, they displayed an increase in the grip to load force, suggesting temporal dyscoordination. Since Vilensky’s study in the 1980s, ASD patients’ walking features have been highly investigated, and a dozen differences were found, such as increased stance duration, reduced stride length, toe striking and increased angular motions [103]. Interestingly, these features are also modified in some ASD mice models. ASD patients seem to slow down their movements in relation to task difficulty and need more practice than typical individuals. These deficits in complex social motor skills could be associated with social and imitation impairments and over-reliance on proprioceptive feedback in motor control and learning (for review, see [104]).

### 2.3. Cellular Correlates

The PC number decreases from 35 to 95% in ASD patients [49,63,75,84], and their soma size and, consequently, their density [29,84], are affected too. Differences in the extent of the reported decreases seem to result from a high variability both in subjects and methods. One of the pioneer post-mortem studies found PC loss and ectopic PCs within the molecular layer only in adults, whereas PC inclusions were described in a sole child case [30]. In parallel, fewer axons were found in PCs from ASD children (male and female, 3.6–13.3 years old) using diffusion MRI tractography [31]. These findings highlight age as a crucial parameter in ASD cellular consequences. The spectrum of symptoms in this disorder might account for the difficulty in obtaining homogenous results in both patients and animal models. This is even the case with genetic mouse models of ASD, where different cellular outcomes have been reported. For instance, PC density was reduced in both Fmr1 KO mice (C57BLJ6J, P30) [44] and mice with a specific deletion of the tuberous sclerosis complex 2 (TSC2) [45], whereas an increase was observed in Mecp2-deficient mice [46]. In our study, Shank3^∆c/∆c^ mice displayed an extensive loss of PC in both crus I and II, but only in males [49]. While gait was disrupted only in males, deficits in social novelty were observed in both sexes. In environmental ASD models, our previous studies showed PC loss in both crus II and 7cb in males and PM in females, with poly I:C prenatal insult [48]. In the VPA mouse model (E12.5; 450 mg/kg) (Table 1), PC number was reduced in crus I in males and crus II in females [47]. This sex-specific reduction was correlated with behavioral tests assessing sociability and motor impairments. Sexual dimorphism was also found in another study that reported PC decrease only in VPA males in the cerebellar lobules VI, VIII, IX and the paramedian one [16]. Clinical studies also established correlations between sex and regional-specific PC loss. For instance, Skefos et al. found that ASD male patients (7–56 years old) exhibited a 21% decrease in regional volume-weighted mean compared to females (4–21 years old). Only males presented a reduction in PC in the lobule X of the flocculonodular lobe. The posterolateral region seems to be the most affected, specifically the lobule VII hemispheres, which are crucial sensorimotor areas with reciprocal interaction with the PFC and the posterior parietal cortex [105] (Table 2).

Differences between males and females in ASD seem to find their origin in the brain masculinization during brain development, a critical period that puts males at risk regarding neurodevelopmental disorders. During the second postnatal week, arachidonic acid and estradiol production peak. It has been shown that inflammation or nonsteroidal anti-inflammatory drugs (NSAIDs) result in impaired play behavior in males [106]. Even though it is still assumed that the cerebellum is not sexually dimorphic, as opposed to the preoptic area, for example, specific subregions such as lobules VI and VII have been demonstrated to represent a particular sexual orientation dimorphism related to emotion and sensation [106]. In the cerebellum, prostaglandins stimulate aromatase and local estradiol production. PCs respond to prostaglandin E2 (PGE2) and 17-estradiol (E2) levels. Indeed, E2 induced BDNF expression at physiological levels and promoted PC dendritic growth, spinogenesis and synaptogenesis during neonatal life. High E2 levels stop dendritic growth and reduce excitatory synapses number [89,106]. Given that E2 is the primary hormone in females, this raises the hypothesis that E2 could protect females from environmental insults, whether toxic, pharmacologic or immune. Indeed, only male rats exposed to LPS or PGE2 during the second postnatal week displayed reduced PC arborization and impaired juvenile social play behavior [107] (Table 1).

Unfortunately, not all studies separate male and female groups; many use only males, and some studies even pool the data from both sexes, undermining sex differences. For instance, rats (male and female pooled) prenatally exposed to a single VPA dose exhibited a reduced number of neurons in the cerebellum, abnormal dendritic branching and reduced density dendritic spines affecting axonal projections of PC [53,54]. Double-dose VPA-exposed rats (male and female pooled, E10 and E12 800 mg/kg) exhibited an increased number of ectopic PCs correlated with a 21% reduction in soma size compared to control ones in all ten vermal lobules. While lobules IV and VIc were both affected in decreased PC soma size (30 and 39%, respectively) and PC number (55 and 36%, respectively), lobule VII seemed to be less affected, with only a 9% decrease. In the same study, VPA gestational exposure induced a reduction in Calbindin across all ten vermal lobules. Only 65% of vermal PC were Calbindin positive compared to control animals that were 90% PC Calbindin positive [50]. In this model, Calbindin-positive PC dendrites were shorter and showed reduced branching complexity, in accordance with a previously reported slight increase in spine length and volume in Calbindin KO mice [85], suggesting that a lack of Calbindin could lead to impaired spine morphology, hence reducing synapse formation. Interestingly, no significant changes were observed in the Parvalbumin KO mice. Nonetheless, double transgenic KO mice for the two EF-hand type Ca^2+^ binding proteins (Calbindin and Parvalbumin (PV)) showed significant differences in PC spine morphology compared to wild type [85]. Synapse establishment is a crucial process for PC to engage in cognitive and motor tasks. It has been known that mutations of genes coding for SHANK are impaired in autistic patients [32]. These mutations lead to fewer mature dendritic spines due to impaired spine induction and morphology. In detail, this leads to reduced mature glutamatergic synapses, which would affect cognitive functions. Studies on the male Fmr1 KO mouse model also showed dendritic impairments in the cerebellum, with PC exhibiting longer and immature dendritic spines [44,88]. The eye-blink conditioning, a behavior managed by the cerebellum and mainly the interpositus nucleus, was impaired in both Fmr1 KO and PC-specific Fmr1 KO mice [88] (Table 1). Although eye-blink conditioning was impaired in FXS patients [88,108], repeated training led to improvements in adult patients [108].

Bergmann cells are cerebellar astrocytes that are crucial for PC dendritic formation and maintenance. Post-mortem analysis of six male ASD patients revealed that these cells were activated and reactive in cerebellar areas where PC were reduced [64,90]. This was also observed in VPA rats [86] and Fmr1 KO mice [44]. The microglia oversee the proper neuronal development and have been demonstrated to be involved in synaptic density, spatial localization, morphology, process retraction and thickening, resulting in synaptic pruning. The co-activation of microglia and astroglia seems to correlate with degenerating PC, granule cells and axons. Indeed, post-mortem analysis of ASD patients’ cerebellum (male and female, 5–44 years old) showed a pattern of high microglial activation compared to control tissues [90]. However, in our study, no change in microglia was found in the poly I:C animals (P45) [48]. Since microglia are known to be activated during a specific time window, a more detailed investigation of the timing of these processes is needed. In addition, microglia and astroglia activation in post-mortem samples could be an independent event that may not be linked to ASD but rather to traumatic cause of death (listed in [90]). PCs seem to be at the center of the gliosis found in ASD patients and animal models through auto-toxic mechanisms, independent of adaptative immunity. Indeed, markers found in the perineural compartment of PCs raise the hypothesis of complement system involvement in immunopathogenic means, similarly to what is reported in neurodegenerative disorders. These findings indicate that cerebellar abnormalities found in ASD patients could be due not only to prenatal developmental insults but also to glia-mediated chronic neuroinflammatory processes persisting throughout life.

Other cerebellar cell types, such as basket and stellate cells, seem to be preserved in ASD patients [109]. These observations reinforce the hypothesis that PC density loss could be due to an insult during late prenatal periods, when PCs and inferior olivary neurons establish their connections [105]. Only one study found a decreased granule cell number in ASD patients [90], but findings in this cellular population rarely constitute a perfect match through animal models of various cerebellar pathologies.

### 2.4. Neurotransmission Systems Implicated

Glutamate and GABA have consistently been reported to be the most affected neurotransmission systems in ASD. The nature of motor and cognitive impairments in ASD points toward an excitatory/inhibitory imbalance within the cerebellum (for review, see [87]). The GABA–glutamate imbalance was thought to be responsible only for seizures occurring in some schizophrenic and ASD patients, but recent evidence has proven a wider array of pathologies [51]. For instance, glutamate has been extensively related to neurogenesis, synaptogenesis and neuronal maintenance in relation with emotional behavior acquisition. The cerebellum contains various AMPA, NMDA and kainate glutamate receptors. Using microarrays, autoradiography and western blot on post-mortem patient samples, Purcell et al. compared markers of glutamatergic neurotransmission in ASD subjects (5–54 years old) and neurotypical ones (2.4–53 years old) and reported a decreased AMPA receptors (AMPAR) density in patients’ cerebellum [110]. Since glutamatergic interacting proteins 1/2 (Grip1/2) regulate AMPAR trafficking and synaptic strength, PC–AMPAR signaling in ASD was recently examined in a PC-specific knockout mouse model. The specific loss of expression of Grip1/2 in PC resulted in an increased repetitive self-grooming in 3-month-old male mice and impaired mGluR long-term depression (LTD) at the parallel fiber (PF)-PC synapses [51]. In addition, mGluR5 and Arc were increased in a possible attempt to compensate for AMPAR inefficient recycling in the absence of Grip 1/2 [51]. However, no PC loss was found in this animal model [109]. Both excitatory amino acid transporter 1 (EAAT1) and 2 (EAAT2) mRNA and proteins were increased in ASD patients [87]. The EAAT1 and 2 are mainly expressed by Bergmann astroglia in the cerebellum, which is highly activated in ASD patients. The glutamatergic extracellular concentrations are thus suspected to be abnormally elevated in ASD subjects, resulting in an imbalance in excitation/inhibition. This may implicate a glutamate-mediated strong activation of PCs, which in turn would lead to cerebellar inhibition. Interestingly, the precursor of the glutamate synthesis, glutamine, is also upregulated in the left cerebellum of ASD patients [52]. However, a few studies did not report any increase in the ASD glutamatergic system. For instance, DeVito et al. used proton magnetic resonance spectroscopic imaging (1H MRSI) to detect various low-molecular-weight metabolites in vivo in young male ASD patients (6–17 years old) and control subjects (6–16 years old) [111]. Among the studied metabolites, both N-acetyl aspartate and glutamine were reduced in the cerebellum of ASD patients compared to controls, suggesting not only neuronal loss or dysfunction but also reduced levels of glutamate. These results suggest widespread reductions in gray matter neuronal integrity and a dysfunction of cerebellar glutamatergic neurons in ASD patients. However, it is to be noted that no females were used in this study, contrary to the work from Hassan et al. [52]. This is of relevance, as the menstrual cycle influences women’s neurotransmitter levels across cortical regions [112]. Further studies with a larger cohort, including sex, age and menstrual cycle parameters, are required to determine parameters influencing the glutamate and glutamine levels in the ASD cerebellum [113]. Glutamatergic transmission is also modified in ASD animal models. Shank2-deficient mice (Shank2^−/−)^ displayed abnormal and repetitive behaviors, as well as autism-like social deficit behaviors [114]. The cerebellar synaptosomes from these mice had fewer AMPA receptor subunits (GluA1 and GluA2) than control without affecting dendritic arborization and postsynaptic density. Electrophysiological recordings in these animals revealed deficits in long-term potentiation (LTP) in PF–PC [114]. In line with these findings, mice with a Shank2 deletion restricted to PC (*Pcp2-Cre;Shank2fl/fl* mice) displayed an interesting phenotype that only partially related to ASD symptomatology [98]. Indeed, social behavior and repetitive behaviors were not observed in this mouse line, as these transgenic mice showed mainly motor coordination impairments and increased anxiety. The PC lacking Shank2 protein (*Pcp2-Cre;Shank2fl/fl* mice) displayed fewer miniature excitatory postsynaptic currents (mEPSC) and fewer GluA1,GluA2,GluN2C, VGluT1 and GluD2 protein levels than control [98]. Specific loss of the TSC1 in the PC (*L7Cre;Tsc1flox/1* and *L7Cre;Tsc1flox/flox*) results in ASD behaviors [96]. Indeed, mice carrying this mutation displayed an increase in stereotypic movements, abnormal behavior and changes in PC electrophysical properties. PC lacking Tsc1 had a decrease in action potential frequency and EPSCs, but not IPSCs. This modification led to a decrease in the Excitation:Inhibition (E:I) ratio in mutant mice compared to control [96].

GABA has been studied to a lesser extent in the cerebellum compared to glutamate. Indeed, even though PC generate a GABAergic output, their cerebellar inputs are mainly glutamatergic, except for interneurons regulating PCs’ firing pattern. Interestingly, Calbindin reduction in mice resulted in abnormal firing patterns in PC, such as decreased complex spike duration and pause and simple high spike firing rate [55]. These findings suggest that GABAergic inputs on the PC could be partially dependent on Calbindin levels. Lower densities of the two GABA receptor subtypes—GABA-A and GABA-B—have been found in ASD individuals. The GABA-A receptor is known for its fast inhibitory action, whereas GABA-B receptor activation results in excitatory/inhibitory regulation. The subunits GABA-A α protein levels and GABA-B R1 receptor density and levels of glutamic acid decarboxylase (GAD) 65 and 67 proteins, in charge of glutamate to GABA conversion, were all found to be decreased in the cerebella of ASD patients [97,115]. Furthermore, ASD children (5–15 years) showed increased GABA concentration in their plasma [116]. GABAρ3 is a subunit of the GABA-A receptor, with a high affinity for GABA, providing the receptor with low desensitization upon activation. The GABAρ3-composed GABA-A receptor plays an important role in regulating GABAergic transmission during the postnatal development of the cerebellum [27]. In the VPA model, GABAρ3 is decreased by 43% in the lobule X [99], in charge of the gaze coordination. Importantly, GABAρ3 level linearly increases during typical development, but not in the VPA model, as it chronically decreases at each studied time point (P4: −54%, P30: −83%) [99]. A significant reduction of GABA-Aβ1 and GABA-Aβ2 levels were also observed in the cerebellum of Fmr1 KO mice. Interestingly, the decrease in GABA-Aβ1 mRNA is only observed in the cerebellum, whereas GABA-Aβ2 mRNA levels also drop in the cortex, hippocampus and diencephalon [117]. In this mouse model, the administration of a GABA-A (Diazepam) or GABA-B (STX209) agonists results in the improvement of several behavioral deficits and a partial rescue at the molecular level [95,118] (Table 2).

Excitatory/inhibitory imbalance is one of the major hypotheses explaining ASD symptoms. Of interest is the finding that both ASD patients and ASD mouse models (FMRP) displayed surprising GABA dysregulations and glutamate receptor subunit changes, even in *post-mortem* cerebella of ASD subjects, without FXS or FMRP being downregulated, which was associated with increased levels of mGluR5 and decreased levels in GABA-A β3 subunits [92]. Fmr1 KO mouse models confirmed these findings, as PSD-95 (Postsynaptic Density protein 95) and mGluR5 were found to be increased in the cerebellum [93,94,119]. In our study, mGluR5 levels were reduced in the cerebellum of Shank3^ΔC/ΔC^ males [49]. These findings suggest that mGluR5 plays a crucial role in the synaptic targeting and postsynaptic assembly of the Shank3 scaffolding complex. Drug therapies targeting this protein could be of interest, as the administration of 3-Cyano-N-(1,3diphenyl-1H-pyrazole-5-yl) benzamide, which increases mGluR5 activity, was shown to alleviate functional and behavioral ASD defects [91]. In another study, the Fmr1 KO mouse model showed abnormal presynaptic vesicle dynamics and increased LTD induction at the PF-PC synapse [88]. Other models, such as Shank2 KO mice and specific patDP/+ mutant mice, also showed impaired LTD associated with dysfunctional intrinsic plasticity [56]. The latter is based on findings in ASD patients with mutations in 15q11-q13, a genetic region involved in GABA A β3 subunits and two other subunit candidate genes for ASD [120]. The patDp/+ mice mutants within the cerebellum exhibited impairments in motor coordination, learning, eye-blink conditioning, along with abnormal climbing fiber elimination of the PCs [56]. In the Shank2 KO mice, an increased irregularity in simple spike PC firing, accompanied by increased inhibition, was only found in the posterior cerebellum, possibly underlying cognitive impairments in this model. mGluR1 receptors are expressed by PC mediating LTD plasticity with parallel fiber [121]. In the postnatal developing cerebellum, the mGluR1 activating pathway is involved in axon pruning [122]. The inactivation of this receptor in mice (mGluR1^−/−^) leads to a lack of motor coordination [123]. Even though the role of these receptors is crucial in glutamatergic transmission in the cerebellum, there is yet no direct evidence of mGlur1 dysfunction in ASD. In conclusion, there is an imbalance of the excitation/inhibition inputs in the cerebellum of ASD animal models and patients, with a significant reduction in GABA amount due to a reduction of its synthesizing enzymes GAD65 and GAD67. This is accompanied by a decrease in GABA receptors’ activity, causing an increase in the activity of mGluR5. The mGluR signaling has been shown to be involved in GABA-A receptor stabilization at the synaptic membrane [124] in a healthy context. This may underly the fact that mGluR dysfunctions in ASD are often linked to GABA-A dysregulations. However, it appears that a reduction of the activity of mGluR5 can also lead to social deficits. Pharmacological treatments in animal models indicate that either activating the GABAergic system or inactivating the mGluR5 receptor may be of interest in managing some of the ASD symptoms where a reduced activity of the GABAergic system is reported. In the case of a decrease in mGluR5 levels or activity, the activation of mGluR5 reduces social deficits by restoring the excitation/inhibition balance.

### 2.5. Evidence from Our Previous Work

We have recently set up a series of studies on environmental and genetic animal models of ASD [47,48,49]. Our choice of the models was based mainly on their reported strong construct and face validity, as they were known to replicate both the etiology of the disease and at least some of its cardinal behavioral symptoms. The predictive value of animal models is hard to achieve, especially when no known treatment for the corresponding illness is available, which is the case for ASD. Our main aims when starting these studies were: (i) to determine whether different animal models of the same pathology would yield a spectrum of behavioral and cellular outcomes, mirroring the large and variable range of ASD symptoms in clinical settings, i.e., whether the nature (motor, social, gait) and the severity of the symptoms are variables depending on the etiology of the disease; (ii) to determine whether we can replicate the sexual dimorphism reported in clinical settings, as ASD affects three times more males than females [2], and again, whether sexual dimorphism is observed whatever the animal model and with the same proportion; (iii) to determine whether motor and gait deficits are observed in all animal models and whether they are correlated with the severity of social deficits. All experiments were performed within a relatively short timeframe (4 years in total), in the same laboratory and animal facility, using the same behavioral and bench equipment and software. For this, we have chosen two environmental animal models: the VPA and the poly I:C, a maternal immune activation (MIA) animal model and a genetic animal model bearing a Shank3 deletion. All three animal models showed major motor and gait deficits that were more pronounced in males than in females, but to a variable extent.

VPA animal models were obtained by injecting the drug i.p. at 400 mg/kg to pregnant females at E12.5 [47,125] (Table 1). In this model, we have found that the male offspring, but not females, expressed severe social deficits. However, both sexes showed motor coordination and gait deficits that were more pronounced in males than in females in their severity and variety. Cellular consequences accompanied these differential social and behavioral phenotypes. We reported a decrease in the number of PC found in the crus I cerebellar subregion in males and in crus II in females. In addition, only males showed a reduction in the number of neurons within the motor cortex. No neuronal decrease was found in the striatal region. Of interest is the finding that the severity of motor and gait disturbances was directly and strongly correlated to deficits in social behavior, as mice that had the most motor coordination deficits were the ones that had major social deficits and the highest decrease in PC within the cerebellar cortices.

In order to induce MIA, pregnant mice received a single i.p. injection of poly I:C (20 mg/kg, a double-stranded RNA analog polyinosinic:polycytidylic acid, which presents strong construct and face validity toward ASD and is the preferred MIA paradigm compared to direct injection of viruses [48,126]. Mice that received a Poly I:C injection at the prenatal age of 12.5, inducing an MIA phenotype, showed less dramatic social and motor behavior alterations than those following VPA injection [48]. Only males showed deficits in social behavior and motor coordination. Of interest is that neither gait nor walking skills were affected in either males or females. A reduced number of PC in the cerebellum was found to be more widespread and within distinct lobules in males than in females.

The Shank3^Δc/Δc^ mice that we have used for our studies are those with C-terminal 508 deletion in the Shank3 gene following a frameshift in exon 21, which includes the homer-binding site in the sterile alpha motif domain. Consequently, there is a partial or total loss of the major naturally occurring isoforms of Shank3 proteins in heterozygotes and homozygotes, respectively [127]. This mutation has a strong construct validity, as it mimics a human mutation, which is not the case for several other Shank3 mutations in mice [127]. In homozygote animals, we reported significant impairments in social novelty preference, stereotyped behavior and gait. These were accompanied by a decreased number of PC in restricted cerebellar sub-regions and decreased cerebellar expression of mGluR5. Heterozygote mice showed impairments only in social novelty preference, grooming and decreased mGluR5 expression, but to a much lesser extent than in homozygote mice. All reported deficits were more pronounced in males than in females (Table 2).

Several elements of conclusion can be drawn for our studies: (i) The severity of ASD phenotypes, whether behavioral, cellular or molecular, varies from one animal model to another. This recapitulates in some way the spectrum of the disease, where variability may be due to its etiology, i.e., to what initially caused it. In our study, the VPA model yielded the most robust and severe phenotypes at all explored behavioral and cellular levels. (ii) Females are globally less affected than males in all the paradigms explored, whatever the treatment or the mutation. This is in line with clinical settings, where ASD is reported to be present three times more in males than in females. Notably, females showed no social deficits but still exhibited motor and gait alterations. In addition, the decrease in the number of PC was found in different sub lobules in females than in males. This suggests that ASD may be expressed differently in relation to sexual dimorphism. Thus, the proportion of affected ASD females may be higher than previously suspected if one also investigates motor and gait behavior. Such behaviors may be of relevance to implement in clinical exploration to help diagnose the disease. (iii) When the phenotypes explored are robust, as with the VPA animal model, the magnitude of social deficits can be correlated to both motor and gait deficits and PC cell number. In line with the previous conclusion, this further suggests that exploring motor and gait deficits may constitute an objective, early and quantitative diagnosis tool in ASD.

## 3. Striatal Involvement in ASD

### 3.1. Anatomical Evidence of Striatal Involvement in ASD

The basal ganglia are a group of subcortical nuclei involved primarily in motor skills. The term “Basal Ganglia” refers to the striatum and the globus pallidus, while the substantia nigra (mesencephalon), the subthalamic nuclei (diencephalon) and the pons are related nuclei [128]. The basal ganglia and their related nuclei can be split into three groups: input nuclei receiving incoming information from different cerebral areas, output nuclei sending basal ganglia information to the thalamus and intrinsic nuclei located in between, playing the role of a relay. The striatum is the largest subcortical structure and the only input nucleus of the basal ganglia (Figure 2). Although it has been remarkably conserved through the evolution of the vertebrate lineage for 530 million years, the striatum enhanced its role from primarily motor relay in amniotes to a complex circuitry capable of motor control (action selection), dealing with emotions and motivational state in mammals [129,130]. In primates and humans, the striatum is formed by the caudate and the putamen, separated by the internal capsule in the dorsal part and the nucleus accumbens in the ventral part. For the rest of the mammalians, the striatum is divided into two parts: the dorsal and ventral striatum. The dorsal striatum receives inputs from the sensory and motor cortices, the insular cortex and the orbital cortex on its lateral part (i.e., on the putamen in primates and humans) and the visual cortex, the anterior cingulate cortex, the ventral hippocampus and the amygdala on its medial part (i.e., on the caudate). The nucleus accumbens receives inputs from prelimbic and infralimbic cortices, the amygdala, the hypothalamus and the ventral hippocampus. This input organization leads to each territory’s specializations from the motor to associative and cognitive mechanisms, from dorsolateral to dorsomedial and ventral striatum [129,130].

In the late 20th century, several studies took an interest in symptoms and brain structures other than the cardinal ones by studying the link between ASD and the cerebellum, as we discussed before. First evidence of the involvement of the striatum in the physiopathology of ASD was obtained from imaging studies [131,132,133]. Studies over the last decades highlighted anatomical differences between ASD and non-ASD subjects regarding the striatum and reported a wide range of modifications and inconsistencies. On the one hand, only the putamen and nucleus accumbens volumes appear to increase by, respectively, 22% and 34% in ASD patient brain compared to age-matched controls. The caudate volume was not significantly different between groups [84]. On the other hand, all the striatum, i.e., the caudate, was found to be more prominent in ASD patient brains than age-matched controls in magnetic resonance imaging studies [134,135,136]. Thus, the larger right caudate volume seems to be positively correlated with repetitive and stereotyped behavior [134,135,136] and negatively correlated with insistence on sameness [135]. Interestingly, neuroanatomical changes of the striatum are also found in animal models of ASD but can be different from one model to another: Neuroligin3 Knock-In (B6/129F2 strain, 15 weeks of age), Shank3^−/−^ (B6/SV129 strain, 22–23 weeks of age) and Cntnap2^−/−^ (C57BL/6J strain, 8–9 weeks of age) transgenic mice, for example, have bigger striatum than control littermates [69], while 16p11^+/−^ mice (B6/129F2 strain 32–35 weeks of age) have a bigger nucleus accumbens [137]. Idiopathic models of ASD, such as the BTBR mice (C57BL/6J strain, 11 weeks of age) show a reduction of their striatal volume [138]. Although no modification of the global striatal volume has been reported in FVB/NJ mice exposed in utero to valproic acid (VPA) (National Laboratory Animal Center, Taipei, Taiwan), the balance between striosomes and the matrix is reported to be impaired at postnatal day 8 (P8) depending on the time of the VPA administration. These two compartments receive different cortical inputs: the matrix compartment receives inputs from the cortical sensorimotor areas, while the striosomal compartment receives inputs from prelimbic and insular cortices. The authors reported that both striosomal and matrix areas were affected and could be linked with both social and motor impairments in ASD, highlighting the heterogeneous nature of the disease [28] (Table 2). These differences align with the inconsistencies reported in humans and reflect the large scale of striatal neuroanatomical changes. More generally, all three parts of the striatum are involved in functions that are altered in one way or another in ASD. The dorsolateral striatum has been linked to motor stereotypies in a large range of species, from rodents [139] to monkeys [140], and to insensitivity to reward devaluation [141]. The dorsomedial striatum has been linked to executive dysfunctions and abnormal reactivity [142], and the nucleus accumbens has been linked to the inability to process and respond to social cues [33].

Taken together, these sets of results indicate that all three striatal parts are affected in ASD patient brains and animal models of ASD, reinforcing the link between the striatum and ASD symptoms.

### 3.2. Cellular Consequences Correlates

Despite increasing their volume, the numerical density of neurons is reported to decrease by 15% in the nucleus accumbens and 13% in the putamen in patients diagnosed with ASD and comorbidities [84]. These findings need to be more thoroughly replicated, as 7 out of 14 patients included in the study (50%) were diagnosed with seizure, and death was seizure related for 5 of them. Similar variations in striatal cellular alterations were also found in ASD animal models. In VPA mice and rats, no drastic striatal neuronal loss has been described. A recent paper strongly suggested that the lack of PV staining observed in the striatum of VPA mouse model should not be attributed to cell loss but rather to a decrease in PV protein contents within the interneurons, as the cell can still be observed using another marker (Vicia Villosa lectin, VVA) [26,143]. Nevertheless, the general cellular organization within the striatum appears to be disturbed, with an impaired aggregation of striosomal cells into cell clusters [28] (see Table 1). This could cause (or enhance) an imbalance between sensorimotor and cognitive inputs on the striatum, leading to social and motor impairments.

Representing more than 95% of the striatal neuronal population, medium spiny neurons (MSNs) are the striatum’s main neuronal population and are the striatum’s only output. They are segregated into at least two subtypes according to their axonal projection patterns: striatonigral MSNs, or direct pathway MSNs, expressing dopamine receptor D1 (RD1) and striatopallidal MSNs, or indirect pathway MSNs, expressing dopamine receptor D2 (RD2) [129,144]. All MSNs share a similar morphology and exhibit similar properties, except for excitability, which appears to be increased in DRD2-expressing MSN [145]. Male and female 5-week-old Shank3^B−/−^ MSNs exhibit increased neuronal complexity, increased total dendritic length and surface area, but a lower spine density compared to Shank3^B+/+^ littermates [146]. At the age of 2–4 months, a reduced MSNs spine density is also found in Shank3^B−/−^mice, in both males and females, only in DRD2-expressing MSNs [147]. This is particularly relevant to human findings, where regional quantification revealed that RD2, but not RD1, was significantly more expressed in both caudate and putamen of 4–20-year-old ASD male patients, with multiple ethnicities and non-comorbidities-related death [148] (Table 2).

The striatal interneurons are responsible for local inputs and regulation, which play a major role in the striatum’s functionality. A conjoint depletion of the PV fast-spiking and cholinergic interneurons in the dorsal striatum leads to autism-like behavior in male mice [155], but not in females, suggesting that interneurons, including cholinergic interneurons, could be involved in ASD physiopathology and could take part in the reported sexual dimorphism in ASD patients and animal models [47,48,49]. The central role of cholinergic interneurons is further highlighted, as they are also involved in other brain pathologies. For instance, they have been directly linked to Tourette’s syndrome or attention deficit hyperactivity disorder [156] and motor stereotypies, as a lesion of these cells in the dorsal striatum significantly prolonged their duration [157,158]. A significant increase in the stereotyped behavior was also found in mice (C57BL/6, 6–21 weeks) with a loss of MeCP2 in somatostatin-expressing interneurons, including in the striatum [159]. Nevertheless, there is no available information regarding the involvement of striatal somatostatin-expressing interneurons in ASD physiopathology.

The striatal PV fast-spiking interneurons, responsible for gamma oscillations and feed-forward inhibition onto MSNs, are important in the excitatory/inhibitory balance [129,160]. The inactivation of the PV gene in mice leads to reduced social interactions, reduced ultrasonic vocalizations and repetitive and stereotyped patterns of behavior [161]. This ASD-like phenotype of PV^−/−^ mice is also directly correlated with brain morphological abnormalities similar to those reported in patients. This is accompanied by functional disturbances, as PV^−/−^ fast-spiking interneurons exhibit a lower excitatory post-synaptic current facilitation, i.e., a lower excitatory synaptic plasticity [161]. Interestingly, these striatal PV fast-spiking interneurons are not only related to ASD-like mechanisms when targeted but are commonly affected in ASD models. PV expression levels are reduced in the striatum of Shank1^−/−^ and Shank3^B−/−^ mice, the two mice models of ASD, without a loss of PV interneurons [149]. Similar findings are reported in VPA mice [26] or Cntnap2^−/−^ mice [143]. Given that PV^−/−^ mice ASD-like phenotype is strongly relevant to human ASD core symptoms and related abnormalities and that PV fast-spiking interneurons are affected in a wide range of ASD models, a PV hypothesis of ASD has been formulated recently [162] (Table 3).

Glial cells, including astrocytes and microglia, are now well known to influence synapse formation and function, as astrocytes, for example, can make contact with multiple neurons and express receptors and ion channels that are also expressed in neurons [163]. The role of astrocytes in ASD physiopathology has been demonstrated by re-expressing Mecp2 in whole-brain astrocytes in Mecp2^−/−^ mice (C57BL/6 strain, 4–8 weeks of age), leading to the recovery of motor symptoms [164]. However, no striatum-specific astroglial changes in Shank3^+/ΔC^ and Cntnap2^−/−^ male mice (C57BL/6J strain, 5–6 months of age) were reported [165]. The overexpression of the eukaryotic translation initiation factor 4E (eIF4E) in macrophages in mice, i.e., in microglia in the central nervous system, leads to elevated protein synthesis and abnormal morphology of microglia, as well as an increase in their size and number within the striatum of both males and females (C57BL/6 strain, 2–6 weeks of age). This overexpression of eIF4E in microglia has no phenotypic consequences in females, but it causes ASD-like phenotype in male mice [166], suggesting that microglial impairment could cause ASD with a sexual polarity. Thus, a significant increase in the size and number of microglial cells in the striatum has been reported in Pten^+/−^ and Fmr1^−/−^ mice [166]. These two models exhibit core ASD features, such as social deficits [167,168] and repetitive behavior [169,170], suggesting that microglia could also be impaired due to ASD.

### 3.3. Neurotransmission Systems Implicated

In the early 2000s, John Rubenstein and Michael Merzenich formulated the excitation/inhibition (E/I) imbalance hypothesis of ASD, suggesting that the physiopathology of ASD and their related comorbidities may reflect a disturbance in such a balance [87]. Even though their work focused only on the cortical networks, it may be easily extended to striatal networks, as the striatum receives major excitatory inputs from cortices areas and major inhibitory inputs from the local interneurons network [142]. Added to the dopaminergic and serotoninergic neuromodulations, this would lead to a balance within the striatum, which is essential for its functioning. Another argument suggests an E/I imbalance in the striatum is found in the numerous genes involved in ASD physiopathology. According to the SFARI database, nearly one-third of genes (68 out of 213) used to model ASD in mice directly concern synaptic establishment, strength and/or transmission or are associated with receptors in pre- and/or post-synaptic compartments. Most of these genes are expressed in the striatum, although not exclusively.

Both ionotropic and metabotropic receptors mediate the glutamatergic responses in the striatum and are differentially expressed throughout the striatal territories at the pre-or post-synaptic compartments. While metabotropic receptors are involved in long-term synaptic plasticity, the ionotropic receptors, i.e., AMPA-R and NMDA-R, are responsible for the neurotransmission itself in the striatum [172,177]. In ASD, alterations of the glutamatergic transmission are reported in several models, such as Nlgn1^−/−^ mice, Ngln3^−/−^ mice, Shank3^−/−^ mice [150] or Shank2^−/−^ rats [151]. In Ngln1^−/−^ mice (C57BL/6 strain, P15-P38), the NMDA/AMPA ratio is reported to be decreased in the dorsal striatum and can be restored to WT levels with the application of D-Cycloserine (DCS), a co-activator of NMDA-R. This rescue of NMDA-R function also lowers the increased grooming behavior observed in Nlgn1^−/−^ mice [178]. Interestingly, in the same mice that are 2–3 weeks old, the NMDA/AMPA ratio is reduced in the direct pathway (DRD1-expressing MSNs) but not in the indirect pathway (DRD2-expressing MSNs) and is directly driven by a decrease in GluN2A-containing NMDA-R currents. However, the strength of the synapse, measured by input/output curves, as well as the short-term plasticity, are not reported to be impaired in KO mice compared to WT littermates [154]. Such a dichotomy between direct and indirect pathway has also been described regarding endocannabinoid (CB)-mediated LTD in a model of selective loss of TSC1 in either DRD1-expressing MSNs or DRD2-expressing MSNs (C57BL/6 strain, P40 to P50) [179]. Here, the authors highlighted impairments of CB-mediated LTD in TSC1^−/−^ DRD1-expressing MNSs but not in DRD2-expressing MSNs. In Fmr1^−/−^ mice (C57BL/6J, males only, adult), a loss of LTD has been described in ventral MSNs, without any CB1-R functional alteration [173]. These findings suggest that impairments of glutamatergic neurotransmission and plasticity are directly linked to the neuromodulation system specificity in ASD.

In another neuroligin model of ASD (R451C-NL3 male mice, B6/J strain, 2 months), a high-frequency stimulation (HFS) protocol failed to induce LTD in R451C-NL3 mice, while it did do so in WT littermates. Furthermore, the application of quinpirole to activate D2 receptors, known to be involved in the establishment of LTD during HFS protocol, did not prevent the induction of LTD in the dorsal striatum of mutated mice [180]. Male and female Shank3B^−/−^ mice (C57BL/6 strain, 5 weeks) exhibited a lower MSNs population spike amplitude than control and reduced miniature AMPA-mediated post-synaptic currents (mEPSCs) amplitude, suggesting a reduction of the post-synaptic response on available synapses [146]. Shank3B^−/−^ MSNs also exhibited a reduced mEPSCs frequency, but there were no defects in the paired-pulse ratio, suggesting that the number of functional synapses is decreased. These findings indicate that cortico-striatal impairments are occurring at the post-synaptic, i.e., striatal disturbances. In another Shank3 mouse model (Shank3^e4–9^ mice, C57BL/6 strain, 3–4 weeks), the NMDA/AMPA ratio significantly decreased at glutamatergic synapses on MSNs from both Shank3^e4–9+/−^ and KO mice compared to WT. Nevertheless, no difference in mEPSCs amplitude of frequency has been highlighted [181]. However, in another mouse model of ASD (eIF4E overexpressing mice, males only, 2–6 months), increased mEPSCs amplitude but not frequency has been highlighted [182]. The authors suggested that the increased expression of this translation initiation factor may lead to exaggerated cap-dependent protein synthesis, such as mGluR5 pathway, as it has been described in the hippocampus of Fmr1^−/y^ mice (C57BL/6J, males only, 3–6 weeks). Considering that no AMPA-mediated disturbances have been reported in several models and that mGluR5 are involved in the potentiation of NMDA response, these findings suggest that the disturbances at glutamatergic synapses are due to decreased NMDA-R function. The mGluR5 receptors are mainly involved in the potentiation of NMDA responses in striatal MSNs and are reported to be abnormally distributed and accumulated in striatal MSNs in Shank3^e4–22−/−^ mice (C57BL/6 strain, 2–5 months). These same mice’s MSNs also exhibit enhanced excitability and a loss of plasticity through impairments of high-frequency stimulation (HFS)-induced LTD. All this information suggests that mGluR5 receptors could be dysfunctional and involved in electrophysiological changes observed at excitatory synapses on striatal MSNs in mice models of ASD. To further test such a hypothesis, the authors proposed a pharmacological enhancement of mGluR5 with 3-cyano-N-(1,3-diphenyl-1H-pyrazole-5-yl)-benzamide (CDPPB) and showed that it restores Shank3^e4–22−/−^ MSNs normal excitability and plasticity, as well as behavioral ASD-like behavior [91]. In Shank3Δ11^−/−^ mice (C57BL/6 strain, 3 months), pharmacological enhancement of mGluR5 with CDPPB leads to a total recovery of NMDA receptor functions, compared to WT littermates [91].

Although GABAergic transmission is crucial for proper striatal function, this transmission is far less studied in an ASD context. The first apparent involvement of the GABAergic transmission in ASD physiopathology is following the knock-out of either GABA-A receptor subunit α5 or β3 in mice that leads to ASD-like phenotype [171,174]. In Chd8^+/−^ mice (C57BL/6 strain, males only, 6–8 weeks), the MSNs of the ventral striatum exhibited decreased miniature inhibitory postsynaptic currents (mIPSC) amplitude, compared to WT littermates, while no changes were observed in mIPSC frequency. The authors also report increased spontaneous excitatory postsynaptic currents (sEPSC) amplitude and frequency but no changes in mEPSCs [175]. This suggests that the observed enhancement of the excitatory transmission is partly allowed by a local decrease in inhibitory transmission. In the NL3-cKO mice (C57BL/6 strain, males only, 4–6 weeks), a reduction in the mIPSC frequency, but not amplitude, is reported in ventral striatum D1-MSNs only. The authors also report no changes in the paired-pulse ratio of evoked IPSCs or short-term plasticity and no changes in excitatory transmission. Added to a significant decrease in inhibition/excitation ratio, measured by the ratio between GABA receptor-mediated inhibition and AMPA receptor-mediated excitation, this suggests that the GABAergic alterations observed in inhibition of D1-MSNs of the ventral striatum could be responsible for a shift of balance between synaptic excitation and inhibition [153]. Little is known regarding the implication of dorsal striatum GABAergic transmission in ASD. To date, the most comprehensive study on that topic concerns the Fmr1^−/−^ mouse model, in which extensive GABAergic disturbances have been reported [183]. In this study, the authors found that dorsal striatal MSNs of Fmr1^−/−^ mice (C57BL/6 strain, 2–3 months) have a higher sIPSCs and mIPSCs frequency than WT littermates, but no changes were found in sIPSCs and mIPSCs amplitude and kinetic properties. In addition, the paired-pulse ratio of evoked IPSCs is lower in Frm1^−/−^ mice than WT littermates at 50 ms interstimulus interval, but not at 80 ms interstimulus interval. Taken together, these findings suggest that alterations may occur in pre-synaptic neurons, but in an action-potential-independent manner.

Post-mortem analysis of striata of ASD young patients revealed an increase in RD2 mRNA within MSNs in both the caudate and putamen using radioisotopic in situ hybridization histochemistry [148]. This suggested a dopaminergic imbalance in favor of the indirect pathways in ASD patients [152]. This may support dopaminergic targeted therapeutics that are currently used to manage ASD-related symptoms, such as irritability or aggression. Indeed, treatments with D2 antagonists result in improved sociality (under risperidone) [184] and a decrease in stereotypy scores (under aripiprazole) [185]. Interestingly, it seems that the direct pathway is only dysfunctional when activated during a task. A study in young ASD adults using PET ([^11^C]raclopride, a D2-antagonist) simultaneously with magnetic resonance scanning during an incentive task revealed a decrease in phasic dopaminergic release in the putamen and caudate nucleus, suggesting an impairment in the learning and goal-directed reinforcement [186]. The dopaminergic content is not extensively studied in ASD due to the result disparateness among animal models, especially regarding the striatal dopaminergic content, tyrosine hydroxylase staining and dopaminergic transporter (DAT) levels (for review, see [176]). Dysfunction in the striatal direct pathway in ASD animal models leads to deficits in social interactions and an increase in grooming [153,187,188]. Optogenetic stimulation of the nigrostriatal pathway (ii) was shown to reduce the preference for the social target in the 3-chamberts test (3-CT) with an increase in rearing but no change in grooming or digging behaviors. The stimulation also caused an increase in p-ERK1/2 and p-CAMKIIα levels in the dorsal caudate nucleus-putamen (CPu) [187]. The indirect pathway seems to be not as affected as the direct pathway in ASD mouse models, but studies on early life stress in D2R^+/−^ heterozygous mice showed that these mice display ASD-like phenotype [188] (Table 3).

Alterations in the dopaminergic and glutamatergic systems have been found in several animal models of ASD. For example, a diminution of DAT level is observed in the BTBR mice and Fmr1 KO mice models [188]. It is also interesting to note that in both models, there is an increase in tyrosine hydroxylase-positive neurons (TH+) coupling with VGlut1+ neurons [188] due to the existence of a strong link between VGlut1 and mGluR5 [189]. The inactivation of adenylyl cyclase 5 (AC5), an effector of the D2R and mGluR3 receptor in the striatum, induces ASD-like behaviors in mice [190]. Interestingly, mGluR3 has an antagonistic effect on mGluR5, mirroring the interrelation between D1R and D2R. In the case of WT mice, either the activation of mGluR5 (with DHPG) or the inactivation of mGluR3 (with LY341495) is sufficient to induce ASD-like behaviors [190], as is the inactivation of either GluA1 or GluN2B.

## 4. Epigenetic Alterations in the Brain of ASD Animal Models

Interactions between specific genes and environmental factors play an essential role in the development of ASD, acting directly or indirectly on the transmission and transduction systems detailed above [191]. Environmental factors can affect gene expression without causing changes in the DNA sequence, but by acting via epigenetic mechanisms: DNA methylation, histone modifications or post-transcriptional regulation by non-coding RNAs. Thus, the exposure to environmental factors can cause changes in the expression of critical genes in a crucial period of embryonic and fetal development, resulting in an increased risk of inducing ASD [192]. This is a significant focus of current research, specifically in deciphering the link between environmental insults that occur very early in life and a pathological phenotype that is only observed later in childhood or even adulthood. Epigenetic modifications may be essential molecular mechanisms that can translate early aggressions into lasting brain pathologies that could even be transmitted to the descendants. The best-characterized epigenetic modification is DNA cytosine methylation [193]. This mechanism is already known to be involved in neurodevelopmental disorders, such as Prader Willy/Angelman or Fragile X syndromes, which are major causes of ASD with intellectual disability that have been known for a long time in humans [194]. Methylated cytosines are mainly localized at CpG islands, corresponding to regions of the genome enriched in GC and essentially present in the promoter regions of genes. The primary known function of cytosine methylation at these islands is to repress transcription of the downstream gene(s) [195]. However, there are many exceptions, and the mechanism by which DNA methylation regulates transcription may be specific to different contexts, such as gene content, locus and time of development. Furthermore, DNA methylation is thought to be more abundant in gene bodies, where it probably plays a primary role in fine alternative splicing and expression regulation [196].

In the mouse prefrontal cortex, significant reconfiguration of the methylome occurs throughout synaptogenesis from fetus to young adult [197]. This is thought to be a critical process in defining the molecular identity of neurons [197]. In humans, DNA methylation alterations in the cortex and anterior cingulate gyrus have been observed in patients with ASD [198]. The differentially methylated genes were primarily related to microglial cell specification and synaptic pruning during brain development [198]. More recently, Nardone et al. refined these observations more selectively on neurons after FACS cell sorting of the prefrontal cortex. In doing so, they identified differences in DNA-specific methylation profiles in neuronal nuclei affecting genes related to synaptic, GABAergic and immune processes [199]. This is the first characterization of neuron-specific DNA methylation changes in ASD and is consistent with observations by Lister et al. in mice that showed a specific distribution of methylation in neurons that appeared very different from other tissues, with as many methylated cytosines outside of GC-rich regions as in CpG islands [197]. Epigenetic changes in specific brain structures other than the cortex have only rarely been reported. Notably, the differences in methylation and acetylation of lysine 27 of histone 3 (H3K27) in the cerebellum have been detected in autistic patients compared to controls [200]. Furthermore, in the developing human prefrontal cortex, the 3D organization of chromatin forms loops around synapse-related genes, allowing the regulation of their expression [201]. The formation of these loops is probably regulated by DNA methylation [196]. Other yet unresolved functions of DNA methylation likely play major roles in brain development and neurodevelopmental disorders such as ASD.

Regarding transmission systems, the GABA and Glutamate pathways have already been linked to epigenetic changes. Indeed, an increase in glutamate concentration was identified in the blood of ASD patients compared to healthy subjects [201]. This increase in glutamate metabolism, which could be associated with ASD in children, could be genetic in origin and under epigenetic control. Similarly, the alterations in synaptic physiology in PV interneurons previously described may be related to a deficiency in genes encoding GABA-related enzymes, which alters neuronal GABA content [202].

Concerning the link between epigenetics and environmental insults known to increase the risk of ASD, Richetto et al. [203] showed that maternal infections mimicked by poly I:C injection result in persistent changes in DNA methylation. These changes are visible in many different genomic regions and are related to the timing of prenatal infection. Indeed, late prenatal infection during pregnancy induced methylation changes in significant genes of GABAergic cell development, while early prenatal infection seemed to affect mainly the WNT signaling pathway, involved in the nervous system development [203]. These epigenetic mechanisms have been suggested to be involved in maintaining the long-term effects of prenatal insults, as well as maintaining the cellular dysfunctions they generate [203,204].

In addition, a growing body of evidence proposes the gut microbiota as a likely key player modulating the effect of environmental factors on brain function. This could occur through changes in the immune system and epigenetic machinery. Changes in microbiota composition have been found in patients with behavioral disorders and autism spectrum disorders [205], adding further complexity to the already multifactorial pathogenic mechanisms associated with neurodevelopmental and psychiatric disorders. Epigenetic changes in ASD could notably be induced by metabolites produced by an abnormal gut microbiota [206].

## 5. Conclusions and Future Direction

Although we better understand the implication of the cerebellum and the striatum in ASD, further studies are needed on at least two aspects. First, more knowledge should be gathered on a widespread population of ASD patients without gender or age bias. This means systematically including women and middle-aged and senior ASD patients that are often left out of clinical studies. The state of the brain and its maturation are both sex- and age-dependent. Second, there is an urgent need to focus on developing effective treatments to alleviate ASD symptoms, and knowledge of cellular and molecular correlates of the disease is key to help in this research avenue. Motor social and cognitive functions are common features between the cerebellum and the striatum, which are only two synapses away from each other. A better and more thorough assessment of the corresponding interconnections may lead to breakthroughs in this field.

## Figures and Tables

**Figure 1 ijms-23-02294-f001:**
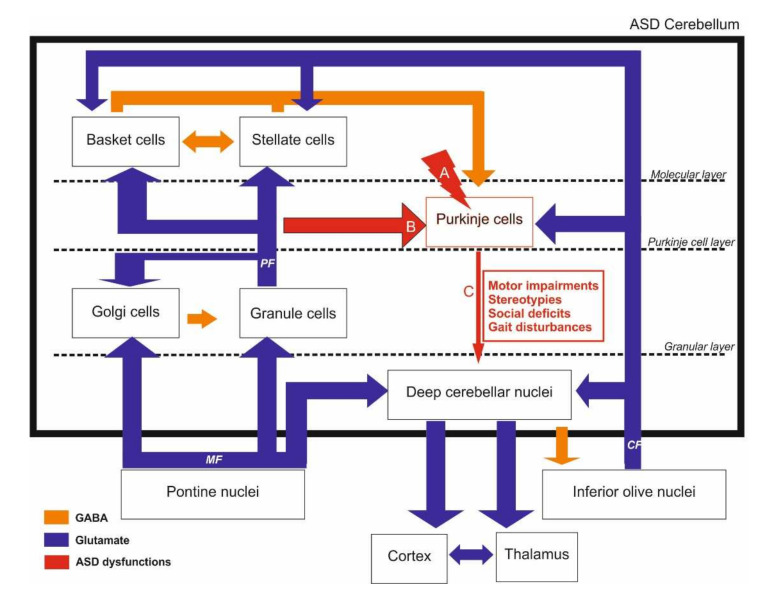
The cerebellum is involved in both motor and social impairments reported in ASD. Dysfunctional Purkinje cells (PC) seem to be at the center of these impairments as they represent the sole output of the cerebellum and receive inputs from both inhibitory and excitatory cells. PC dysfunctions are reported in ASD clinical settings and in animal models (A, B, C). PC intrinsic changes (A) such as reduced PC density was shown in Fmr1 KO mice [44], TSC2f/- mice [45], Mecp2 deficient mice [46], VPA mice [47], polyI:C mice [48] and Shank3ΔC/ΔC mice [49], with regional lobular differences between males and females. Low PC numbers associated with a decrease in soma size and an increase in ectopic PC number were also reported in VPA rats [50], especially in lobule VII hemispheres (crus I and II) that are involved in sensorimotricity. Impaired inputs from granule cells through parallel fibers (B) were also reported. For instance, the mGluR-long term depression (LTD) was altered at the PF-PC synapse in both PC-specific Grip1/2 KO mice and Fmr1 KO mice [51,52]). Furthermore, PC abnormal dendritic branching and reduced density of dendritic spines were found to impair synapse formation in VPA rats [53,54] and in Fmr1 KO mice [44,51]. Outputs from PC onto deep cerebellar nuclei (C) are also impacted, as PC firing pattern is impaired in mice lacking Calbindin, with decreased complex spike duration and pause, as well as decreased simple high spike firing rate [55]. In the Shank2 KO mice, an increased irregularity in simple spike PC firing accompanied by increased inhibition was only found in the posterior cerebellum [56]. Altogether, these PC-focused alterations would lead to dysfunctional cerebellar loop up to the thalamus and cortex leading to both motor and cognitive impairments. MF: Mossy fibers, CF: climbing fibers, PF: parallel fibers.

**Figure 2 ijms-23-02294-f002:**
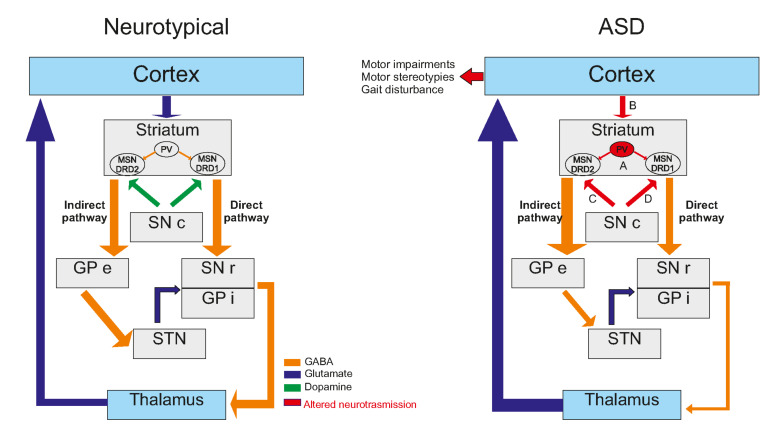
Basal ganglia modifications in ASD. Basal ganglia network is shown to be deeply affected in ASD models. The main alterations are found at 4 points of interest (A,B,C,D). Striatal local inhibition mediated by PV interneurons. (A) is altered in some ASD models like Shank1^−/−^ and Shank3B^−/−^ mice [149], VPA mice [26] or Cntnap2^−/−^ mice [143]. Responsible of feed-forward inhibition, alteration of these interneurons may lead to altered MSNs’ functionality. Interestingly, corticostriatal pathway (B) is widely described as altered in ASD models. For instance, in Ngln^−/−^ mice, Ngln3^−/−^ mice, Shank3^−/−^ mice [150] or Shank2^−/−^ rats [151], alterations of the glutamatergic transmission onto MSNs are reported. Thus, alterations of plasticity were also highlighted in Shank3e4-22^−/−^ mice [91]. However, the dichotomy between direct and indirect pathway allows to highlight some pathway-specific alteration. On the one hand, some alterations were found to be specific of the indirect pathway (C), like an increase of DRD2 expression in the striatum, leading to a dopaminergic imbalance in favor of the indirect pathway [148,152]. On the other hand, some alterations were found to be specific to the direct pathway (D) as alteration of synaptic transmission has been highlighted in DRD1 MSNs but not DRD2 in NL3-cKO mice [153]. Thus, in Nlgn1^−/−^ mice, the AMPA/NMDA ratio was found to be reduced specifically in direct pathway MSNs [154], leading to an altered response.

**Table 1 ijms-23-02294-t001:** Various outcomes in various animal models of valproic acid administration during development in relation with species, dose and age. This table recapitulates the different VPA models and their phenotypes. IP: Intraperitoneal injection, USV: Ultrasonic vocalizations, D1R and D2R: D1 dopamine receptor and D2 dopamine receptor, PC: Purkinje cells, MOR: Mu opioid receptor, PV: Parvalbumin interneurons, mPFC: median Prefrontal cortex, GABAP3: GABA receptor subunit p3.

Species	Treated Animal	Periodicity	Dose	Age of Treatment	Phenotype
Xenopus [13]	Embryo	24 h exposition of the eggs	1 mM2.5 mM5 mM	Stage 8 embryo	5 mM: loss of anterior structures and shortening of anterior-posterior axis in 88% of embryos
C57BL/6J mice [25]	Pregnant female	One IP injection	450 mg/kg	E12.5	 Eye-opening delay  Time to climb the wire and the grid  Immobility in males  Crossing in males  Grooming duration  Rearing in males  Sociability index in males  Errors during challenging beam test  Gait abnormality  PC number in Crus 1/2 and M1/M2
C57BL/6J mice [16]	Pregnant female	One IP injection	600 mg/kg	E12	 PC density in young mice at P13  PC number in males at P40
C57BL/6J mice [26]	Pregnant female	One oral administration	600 mg/kg	E12	 PV expression level but not PV neurons  mPFC volume  KCNC1 mRNA level in forebrain  Kv3.1b mRNA level in forebrain  HCN1 mRNA level in forebrain
CD-1 and GFAP-eGFP mice [27]	Pregnant female	One IP injection	500 mg/kg	E12.5	 CB+ cells CB intensity and GABAP3 intensity in lobule X at P8  Expression of GABP3 in ependymal glial cells  Disrupted GABAP3 expression through development
FVB/NJ mice [28]	Pregnant female	One IP injection	400 mg/kg	E11.5 or E12.75	E11.5  MOR1 expression in caudal striatum at P14  Abnormal cell aggregation into striosomal patch E12.75  MOR1 expression in caudal striatum at P14  CB number rich matrix in rostral in the ventro-medial striatum at P14  Abnormal cell aggregation into striosomal patch  FOXP2+ cell in layer 5 ( S1 cortex) and layer 6 (S2 cortex)  Corticostriatal synapse in rostral striatum  Number, duration, peak frequency and peak amplitude in USV
Long Evans rats [29]	Pregnant female	One IP injection	600 mg/kg	E12.5	 PC number in vermis, anterior, posterior lobes  Granule layer volume in vermis, anterior, posterior lobes
Long Evans rats [30]	Pregnant female	One oral administration	800 mg/kg	E12	 Brain weight  Errors in T-maze  Deficits in females in skilled reaching but improvement in male  Time in open arms in elevated plus maze  Dendritic branching in male  Dendritic length; decrease spine density, decrease cortical thickness
Sprague-Dawley rats [31]	Pregnant female	Two oral administrations	800 mg/kg	E10 and E12	 Body weight between P21 and P28,  brain weight  Body length  Purkinje cell size in all ten lobules  Number of calbindin positive PC in all ten vermal lobules  Primary dendrite thickness.  Time to righting  Length in limb stride  Time to complete static beam and number of errors and of failed attempts
Winstar rats [32]	Young rat	Daily intragastric administration	200 mg/kg	1 month3 months6 months9 months12 months	 Swelling of perikarya and dendritic process  Abnormalities of mitochondria9 and 12 months: severe degenerative changes  Swollen Bergmann astrocytes
Winstar rats [33]	Pregnant female	One IP injection	500 mg/kg	E12.5	 Social play behavior and sociability in 3-CT  Social discrimination ability  D2R expression in Nucleus accumbens  D1R expression in Nucleus accumbens  Resting potential in MSNs  Altered action potential discharge pattern in MSN

**Table 2 ijms-23-02294-t002:** Cerebellar dysfunctions in ASD. This table recapitulates cerebellar dysfunctions found in patients and animal models.

	Children	Teenagers	Adults	Animal Models
<5 years old	5 to 14 years old	14 to 21 years old	>21 years old	
**Anatomical impairment**	**Global cerebellar volume**	N/A	N/A	N/A	Hypoplasia [55,56]	No consistent changes [57]  Fmr1KO [64] and Nlgn4KO^−^ [65],= Shank3^∆C/∆C^, VPA and Poly I:C models [25,66,67]
**White matter (WM) changes**	 WM [68,69]	No change [69]Local thickening in boys [70,71]	No change [69]	N/A	N/A
**Connectivity**	N/A	 in the right Crus I and left inferior parietal lobule [76]	 in right Crus I and left mPFC [77]	 in at least 30 mouse lines for Crus I/II projections [78]  PC firing in left mPFC when right crus I is inhibited [76]
 Right during motor task [72]  in the left supplementary motor areas [72] Global right overconnectivity [73]	
**Cellular correlates**	**Purkinje cell (PC)**	 axon numbers [79]	 axon numbers [79]	 PC density [80]	 PC density [51,80,81,82]  soma size & ectopic PC in molecular layer [81,82]	 PC density [25,66,67,83]  PC density [63]  PC arborization in LPS rats [84]  Dendritic spine density in VPA rats [30]  Increased ectopic PC in VPA rats [29,30]
**Bergmann cells**	N/A	N/A	N/A	Activated/reactivated in PC loss area [51,85]	Activated /reactive in PC loss area in Fmr1^−/−^ [64] and VPA rat [32]
**Microglia**	N/A	High global microglial activation [85]	No changes in poly I:C mice [67]
**Neurotransmission**	**Glutamate**	N/A	 AMPA-R density [86]  glutamate & glutamate metabolite levels [87]	 AMPA-R density [86]  glutamate & its metabolite levels [87]	 AMPA-R density [86]  levels of EAAT1/2 [88]	Impaired mGluR LTD [89,90,91]  mGluR5 expression in Grip1/2-PC, Fmr1^−/−^ and Shank2^−/−^ [90,92,93,94]  mGluR5 expression in Shank3^∆c/∆c^ [66]
**GABA**	 GABA-A β3 level [95]	 GABA-A β3 level [95]	 GABA-A β3 level [95]	 GABA-A α, GABA-B R1 density, GAD65/67 levels [96,97]  GABA-A β3 level [95]	Abnormal PC firing pattern in Calbindin deficient mice [98]  GABA-A ρ3 levels in VPA model [27]  GABA-A β1 & β2 levels in Fmr1 KO [99]

N/A: not applicable.

**Table 3 ijms-23-02294-t003:** Striatal dysfunctions in ASD. This table sums up the striatal dysfunctions identified in patients and animal models of ASD.

	Children	Teenagers	Adults	Animal Models
	<5 yo	5 to 14 yo	14 to 21 yo	>21 yo	
**Anatomical impairment**	** *Volume* **	N/A	 Volume of putamen and nucleus accumbens but not of the caudate [82]	 Striatal volume in cntnap2 ko mice [57]  Striatal volume in btbr mice [137]
N/A	 Volume of putamen, accumbens & caudate [133,134,135,136]	
** *Matrice/striosome organization* **	N/A	N/A	N/A	Impaired balance in vpa exposed mice [28]
**Cellular correlates**	** *Medium spiny neurons* **	N/A	Decreased neuronal density in nucleus accumbens and putamen [82]	 Neuronal complexity,  dendritic length and surface area,  Spine density in Shank3B KO mice [146]  Spine density in DRD2-expressing msns in Shank3B^−/−^ mice [147]
** *PV interneurons* **	N/A	 ePSCs [155]  PV expression [26,143]No cell loss [143,144,145,146,147,148,149,150,151,152,153,154,155,156]
** *Astrocyte* **	N/A	N/a	No striatum-specific changes in shank3^+/δc^ or cntnap2 ko mice [160]
** *Microglia* **	N/A	N/a	 Size and number of microglial cells [161]
**Neurotransmission**	** *Glutamate* **	N/A	N/A	N/A	N/A	 sEPSCs amplitude and frequency [171]  NMDA/AMPA ratio [170] &  NMDA-R function [172]  AMPA mediated mEPSCs amplitude & frequency, no defect in paired-pulse ratio [146]  NMDA/AMPA ratio but no changes in mEPSCs amplitude or frequency [173]Abnormal distribution and  expression of mGluR5 in MSNs in [119]
** *GABA* **	N/A	N/A	N/A	N/A	 mIPSCs amplitude [171]  mIPSCs frequency in ventral DRD1-expressing MSNs only [174]  sIPSCs & mIPSCs frequency in Fmr1 KO mice [175]
** *Dopamine* **	N/A	N/A	 DRD2 mRNA in MSN [148]	N/A	 DAT level in BTBR and Fmr1 KO mice [176]

N/A: not applicable.

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
