# Peer review of "Cerebellar and Striatal Implications in Autism Spectrum Disorders: From Clinical Observations to Animal Models"

_ijms, 2022, doi:10.3390/ijms23042294_

Round 1
Reviewer 1 Report
This review by Thabault et al. is excellent. Drawing on their own work as well as that of other laboratories, the authors describe observations in the cerebellum and striatum of ASD patients and animal models, with a fresh look at the importance of sexual dimorphism. In addition to being well referenced, the review is well written and adds much to the current knowledge of ASD for the scientific community.
I provide below few minor comments:
1- If possible, I would like to push the authors to add schematics in paragraphs 2.1 and 3.1, just to define the basis of the structure we are talking about (with different layers, input and output connections ...). Words are fine, but schematics would help to better follow the description provided. I would also be happy if the authors could use this figure in the rest of the text to highlight the changes that occur in ASD.
2- When talking about animal models. Given the wide variety of animal models (even within a specific model like Shank3), I would encourage authors to be more specific in describing the exact model used or referenced. For example: “In our hands, three ASD mice models (Shank3(ΔC/ΔC) [56], VPA [57] 197 and polyinosinic:polycytidylic acid (poly I:C) [58]) displayed no global changes in cere-198 bellum size although they did show PC loss predominantly in crus I and II”.
3- VPA models. Different concentrations and times of administration are commonly used in the literature. Can the authors describe in a few lines the rationale for using one concentration or another instead of just describing the results obtained?
4- Line 606. This is a strong statement and not completely true. The authors should moderate their statement. See: Prenatal Valproate Exposure Differentially Affects Parvalbumin-Expressing Neurons and Related Circuits in the Cortex and Striatum of Mice from Lauber et al., 2016.
Author Response
We are pleased to read that this reviewer finds our work to be “excellent” providing a « fresh look » and being « well referenced ». We have taken into account all his minor comments and modified the text accordingly.
- We have also added 2 figures and 1 table as requested detailing the cerebellum and striatum networks as well as a table compiling VPA procedures and outputs.
- We have added more specific about the model used (genetic details or dose)
- Line 606 has been modified and now reads :
“Similar variations in striatal cellular alterations were also found in ASD animal models. In VPA mice and rats, no drastic striatal neuronal loss has been described. A recent paper strongly suggested that the lack of PV staining observed in the striatum of VPA mouse model should not be attributed to a cell loss but rather a decrease of PV protein contents within the interneurons, as the cell can still be observed using another marker (Vicia Villosa lectin, VVA) [143,144]. Still, the general cellular organization within the striatum appears to be disturbed, with an impaired aggregation of striosomal cells into cell clusters[137] (see table 1)”
Please find in attached file the new version of the manuscript with 3 tables and 2 figures.
Best

Reviewer 2 Report
The review is interesting and well written, but the authors should have considered other animal models of autism as well.
For example, the authors do not discuss the review of Morowa M. " Autistic-Like Traits in Laboratory Rodents Exposed to Phthalic Acid
Esters During Early Development – an Animal Model of Autism? "of 2021 or that by Takumi T. et al "Behavioral neuroscience of autism" of 2020 or the review of Bruchhage M.K. et al of 2018 "Cerebellar involvement in autism and ADHD" or cite the article of "Bossu JL. et Roux S." The valproate model of autism of 2019.
The bibliography presents many references in which the year of publication is written in bold and others not
Author Response
We are pleased to read that this reviewer finds our work interesting and a significant contribution to the field. We have taken into account his minor comments and modified the text accordingly
- We acknowledge other ASD animal models in the review but we could not discussed all of them.
We have added this sentence “We will focus on the VPA as an environmental model of ASD in this review event though other have been described as the phthalic acid ester exposition (for review [210]) .”
- We have now also added a detailed table recapitulating the VPA various models in relation with dose, time of administration and corresponding phenotype in a table and the Roux’s article is cited [17]
- We have corrected the reference issues you mentioned
Please find in attached file the new version of the manuscript with 3 tables and 2 figures.
Best

Reviewer 3 Report
This review article elaborated evidence indicating critical roles of the cerebellum and the striatum in the pathophysiology of autism. The authors demonstrated evidence ranging from clinical observation such as post-mortem or neuroimaging findings to anatomical, cellular, and physiological findings in animal models, then discussed how consistent or contradictory those findings are between species, genetic or environmental etiologies, and sexes. Historical backgrounds why and how these two particular motor-related brain regions attracted researchers’ attention in autism research were also discussed, which helps readers better understand the importance of striatal and cerebellar studies. Three articles from the authors’ laboratory were also introduced and what insights their own findings added to the field in the context of autism research. The article was well written, the logic was clear, and the interpretation of previous findings was appropriate, which therefore will serve as a good review article for future researchers in the field.
Here I have several comments and suggestions.
Line 58-59 :
Numbers of evidence indicate that the integrity of blood-brain barrier in developing brains including fetus ones is similar to the one in adult brains. This sentence may need to be reconsidered.
For instance, please see these reviews
DOI: 10.1016/j.neuro.2011.12.009
DOI: 10.3389/fnins.2014.00404
DOI: 10.1113/JP274938
Line 68-69 and Line 71:
There are also controversies and criticisms against social interaction and motor stereotypy tests and USVs. This sentence may need to be reconsidered.
Please see these reviews, for example
DOI: 10.1016/j.neuroscience.2020.05.010
DOI: 10.1159/000330213
Line 174 and Line 196-197 :
It is not clear how PC loss leads to reduced cerebellar volume or cerebellar hypoplasia. These statements could be too strong. Please cite articles strongly suggesting the causal relationship if any.
Line 338 :
Earlier works also examined eye-blink conditioning in FXS patients part of which showed contradictory results.
DOI: 10.1037//0894-4105.8.1.14
DOI: 10.1037/a0015662
Line 355-358 :
Please cite related articles. Maybe these are relevant.
DOI: 10.1016/j.smim.2019.101340
DOI: 10.3389/fnins.2020.00023
Line 368:
There are numbers of relevant studies which the authors may need to discuss in this section. The authors introduced findings from several studies in detail, but the reason why they focused on these papers is not fully clear. I would list here important related articles. Please consider.
DOI: 10.1016/j.biopsych.2009.12.022
DOI: 10.1038/nature11310
DOI: 10.1038/s41380-018-0018-4
DOI: 10.1111/ejn.13051
DOI: 10.1038/ncomms12627
DOI: 10.1523/JNEUROSCI.1849-16.2016
DOI: 10.1523/JNEUROSCI.1356-16.2016
DOI: 10.3389/fncir.2021.676891
DOI: 10.1016/j.nbd.2018.08.026
DOI: 10.1016/j.neuron.2015.07.020
DOI: 10.1038/npp.2015.339
DOI: 10.1016/j.isci.2020.101820
DOI: 10.1016/j.celrep.2019.01.004
DOI: 10.1016/j.celrep.2020.107703
DOI: 10.7554/eLife.46773
DOI: 10.1093/brain/awaa028
DOI: 10.1016/j.celrep.2021.108932
DOI: 10.1073/pnas.1809382115
DOI: 10.1371/journal.pone.0099524
DOI: 10.1523/JNEUROSCI.2279-06.2007
DOI: 10.1038/s41380-018-0240-0
There are also review articles discussing related topics.
DOI: 10.3389/fnins.2015.00420
DOI: 10.1155/2017/6595740
DOI: 10.1016/j.conb.2017.12.016
Line 435:
The theme of this paragraph is not very clear. E/I imbalance in the leading sentence is not really discussed, rather mGluR dependent plasticity is discussed in detail. Also, mGluR1 instead of mGluR5 is a dominant isoform of Gp1 mGluRs in the cerebellum whose activity is known to be required for cerebellar synaptic plasticity (DOI: 10.1007/BF02941886 ; DOI: 10.12688/f1000research.10485.1). Ref 107 studied mGluR signaling in cortical and hippocampal neurons in Fmr1 KO.
Line 461:
The rationale why a decrease in GABA receptors’ activity causes an increase in mGluR activity is not clear. Please clarify.
Line 462-466:
Please cite corresponding articles. Also, recent work suggested that genetic reduction of mGluR5 activity causes a deficit in social behavior in mice.
DOI: 10.1016/j.bbr.2021.113378
Line 657:
Same as section 2.4. These papers, for instance, reported important findings. Please consider.
DOI: 10.1038/npp.2010.19
DOI: 10.1038/ncomms2045
DOI: 10.1021/acschemneuro.7b00398
DOI: 10.1038/s41551-018-0252-8
DOI: 10.1016/j.celrep.2021.109511
DOI: 10.1016/j.nbd.2020.104746
DOI: 10.1038/nature09965
DOI: 10.1038/nature16971
DOI: 10.1038/nn.4260
DOI: 10.1038/ncomms11459
DOI: 10.1016/j.celrep.2019.10.021
DOI: 10.1016/j.biopsych.2019.03.974
DOI: 10.1073/pnas.1719014115
DOI: 10.1038/nn.4380
DOI: 10.1038/nature11782
DOI: 10.1523/JNEUROSCI.2684-17.2018
Review article is also available.
DOI: 10.1002/jnr.24560
Minor point
Line66 and Line 472 : “face validity” might be more appropriate instead of “face value”?
Line 90 : Cartoon representation of the cerebellar circuit summarizing subsection 2.1 may make the article more intuitive and instructive. And maybe in section 3, the striatal circuit as well.
Line 143-145: Climbing fibers also pass through the granule cell layers
Line 150 : “both projections” may be typo. Parallel fibers are not originated from IO.
Line 202 : “increase” may need to be replaced to “increase in white matter volume” or so for more clarity.
Line 330, 488, 777, 779, 786: Please indicate the full names for PV, MIA, 3-CT, CPu, TH
Line 733-734 : “glutamatergic” may be a typo for “GABAergic”
Line 848 : “198” may be a typo for “190”
Author Response
We would like to thank that reviewer for his/her comments. We have taken in consideration numerous revisions :
- Line 58-59 :
Our sentence was too reductive, we took in consideration your comment as you can see below
“There is not a consensus about how the cytokines could reach the fetus brain. They could reach it by disrupting the brain blood barrier(BBB) ( Mogami 2018) or by the actions of proinflammatory cytokines able to cross the BBB (for review Zawadzka 2021). »
- Line 68-69 and Line 71:
We have added nuances to the sentence , also the USV ambiguities were already mentioned
“while social interaction and motor stereotypy can robustly be observed and scored in rodents using various and complementary behavioral tests, cognitive stereotypy and intellectual disability cannot be adequately assessed »
“Thus, it seems challenging to observe consistent and robust USV changes through different ASD mouse models. »
- Line 174 and Line 196-197 :
It is not clear how PC loss leads to reduced cerebellar volume or cerebellar hypoplasia. These statements could be too strong. Please cite articles strongly suggesting the causal relationship if any.
Response: We agree with the reviewer’s comment; our original statements were too strong and based on paper discussions (Murakami et al., 1989); the causality link is yet to be determined. We have now changed our sentences accordingly that now reads:
- Line 174
“Indeed, there are three cerebellar-related deficits reported in ASD patients based on imaging and post-mortem histological observations: (i) a decrease in the number of PC, (ii) reduced cerebellar volume and (iii) disrupted circuitry between the cerebellum and connecting brain areas such as the thalamus, the pons and the cortex [38,39].”
- Line 196-197
“PC loss has been consistently described in ASD patient brains and cerebellar hypoplasia was found in most cases [38, 39,54,55].”
- Line 338 :
" Although eye-blink conditioning was impaired in FXS patients [87,88], repeated training lead to improvements in adult patients [88] ."
- Line 368:
We thanks the reviewer for these interesting papers. We have added a paragraph about the cerebellar neurotransmission in ASD animal models using some of these references adding value to the 2.4 segment.
“The glutamatergic transmission is also modified in ASD animal models. Shank2 defi-cient mice (Shank2 -/-) displayed abnormal and repetitive behaviors, as well as au-tism-like social deficit behaviors [99]. The cerebellar synaptosomes from these mice had fewer AMPA receptors subunits (GluA1 and GluA2) than control without affect-ing dendritic arborization and postsynaptic density. Electrophysiological recordings in these animals revealed deficits in Long term potentiation (LTP) inPF-PC [99]. In line with these findings, mice with a Shank2 deletion restricted to PC (Pcp2-Cre;Shank2fl/fl mice) displayed an interesting phenotype that only partially related to ASD sympto-matology [100]. Indeed, social behavior and repetitive behaviors were not observed in this mouse line , as these transgenic mice showed mainly motor coordination im-pairments and increased anxiety. The PC lacking Shank2 protein (Pcp2-Cre;Shank2fl/fl mice) displayed fewer miniature excitatory postsynaptic currents (mEPSC) and fewer GluA1,GluA2,GluN2C, VGluT1 and GluD2 protein levels than control [100]. Specific loss of the TSC1 in the PC ((L7Cre;Tsc1flox/1 and L7Cre;Tsc1flox/flox)) results in ASD behaviors [101]. Indeed, mice carrying this mutation displayed an increase in stereo-typic movements, abnormal behavior and changes in PC electrophysical properties. PC lacking Tsc1 had a decrease in action potential frequency and EPSCs, but not IPSCs. This modification led to a decrease in the Excitation:Inhibition (E:I) ratio in mutant mice compared to control [101].”
- Line 435:
“mGluR1 receptors are expressed by PC mediating LTD plasticity with parallel fiber [118] . In the postnatal developing cerebellum, the mGluR1 activating pathway is in-volved in axon pruning [119]. Inactivation of this receptor in mouse (mGluR1 -/-) leads to a lack of motor coordination [120]. Even though the f role of these receptors are cru-cial in glutamatergic transmission in the cerebellum, , there are yey no direct evi-dences of mGlur1 dysfunctions in ASD .”
- Line 461:
“The mGluR signaling has been shown to be involved in the GABA-A receptor stabilization at the synaptic membrane [121] in a healthy context. This may underly the fact that mGluR dysfunctions in ASD are often linked to GABA-A dysregulations”.
- Line 462-466:
The paper is really interesting but as the mutation is affecting the whole brain, it is difficult to decipher the specific involvement of mGluR5 in the cerebellum or the striatum.
Line 657:
“Such a dichotomy between direct and indirect pathway has also been described regarding endocannabinoid (CB) mediated LTD in a model of selective loss of TSC1 in either DRD1-expressing MSNs or DRD2-expressing MSNs (C57BL/6 strain, P40 to P50) (Benthall et al., 2021). Authors highlighted here impairments of CB mediated LTD in TSC1-/- DRD1-expressing MNSs but not in DRD2-expressing MSNs. In Fmr1-/- mice (C57BL/6J, males only, adult), a loss of LTD has been described in ventral MSNs, without any CB1-R functional alteration (Jung et al., 2012). These findings suggest that impairments of glutamatergic neurotransmission and plasticity are directly linked to the neuromodulation system specificity in ASD."
"However, in another mouse model of ASD (eIF4E overexpressing mice, males only, 2-6 months), increased mEPSCs amplitude but not frequency has been highlighted (Santini et al., 2013). The authors suggested that the increased expression of this translation initiation factor may lead to exaggerated cap-dependent protein synthesis, such as mGluR5 pathway as it has been described in the hippocampus of Fmr1-/y mice (C57BL/6J, males only, 3-6 weeks). Considering that no AMPA-mediated disturbances has been reported in several models and that mGluR5 are involved in the potentiation of NMDA response, these findings suggest that the disturbances at glutamatergic synapses are due to decreased NMDA-R function“
Minor point
- Line66 and Line 472 : “face validity” might be more appropriate instead of “face value”?
Response: Done
- Line 90 : Cartoon representation of the cerebellar circuit summarizing subsection 2.1 may make the article more intuitive and instructive. And maybe in section 3, the striatal circuit as well.
Two figures have been provided about the cerebellum (figure 1) and the striatum (figure 2).
- Line 143-145: Climbing fibers also pass through the granule cell layers
Response: Done
- Line 150 : “both projections” may be typo. Parallel fibers are not originated from IO.
Response: Done
- Line 202 : “increase” may need to be replaced to “increase in white matter volume” or so for more clarity.
Response: Done
- Line 330, 488, 777, 779, 786: Please indicate the full names for PV, MIA, 3-CT, CPu, TH
Response: Done
- Line 733-734 : “glutamatergic” may be a typo for “GABAergic”
Response: Done
- Line 848 : “198” may be a typo for “190”
Response: Done
Please find in attached file the new version of the manuscript with 3 tables and 2 figures.
